# Global identification and mapping of socio-ecological production landscapes with the Satoyama Index

Yoji Natori[1]¤*, Akihiko Hino[2]

**1** Conservation International Japan, Tokyo, Japan, **2** EnVision Conservation Office, Hokkaido, Japan

¤ Current address: Faculty of International Liberal Arts, Akita International University, Akita, Japan
* ynatori@aiu.ac.jp

## Abstract

Production landscapes play an important role in conserving biodiversity outside protected areas. Socio-ecological production landscapes (SEPL) are places where people use for primary production that conserve biodiversity. Such places can be found around the world, but a lack of geographic information on SEPL has resulted in their potential for conservation being neglected in policies and programs. We tested the global applicability of the Satoyama Index for identifying SEPL in multi-use cultural landscapes using global land use/cover data and two datasets of known SEPL. We found that the Satoyama Index, which was developed with a focus on biodiversity and tested in Japan, could be used globally to identify landscapes resulting from complex interactions between people and nature with statistical significance. This makes SEPL more relevant in the global conservation discourse. As the Satoyama Index mapping revealed that approximately 80% of SEPL occur outside recognized conservation priorities, such as protected areas and key biodiversity areas, identifying SEPL under the scheme of other area-based conservation measures (OECM) may bring more conservation attention to SEPL. Based on the issues identified in the SEPL mapping, we discuss ways that could improve the Satoyama Index mapping at global scale with the longitudinal temporal dimension and at more local scale with spatial and thematic resolution.

## Introduction

With land use—particularly, agriculture—as the major driver of biodiversity loss [1, 2], and given that much biodiversity resides outside existing protected areas [3, 4], the management of production landscapes is of primary conservation importance. The world is in the sixth mass extinction episode [5–7] and in a state beyond the planetary boundary [8]. With the increased need for food production due to a growing population, agricultural development is on the rise [9]. Under the United Nations Sustainable Development Goals, trade-offs between poverty and hunger eradication goals and conservation goals could further accelerate this pressure. The development of renewable energy infrastructure as a means to address climate change has

**Data Availability Statement:** All relevant data are within the paper and its Supporting Information files.

**Funding:** This study was funded by the Global Environment Facility under the "GEF-Satoyama

Project" (Project ID: 5784; http://www.thegef.org/projects) executed by Conservation International Japan. English proofreading was supported by Akita International University. There was no additional external funding received for this study. The funders had no role in study design, data collection and analysis, decision to publish, or preparation of the manuscript.

**Competing interests:** The authors have declared that no competing interests exist.

been degrading the integrity of known sites of biodiversity importance, such as protected areas, key biodiversity areas (KBAs), and remaining wilderness in Western Europe, and the same is also expected in Southeast Asia in the near future [10]. Sites of potential importance that have yet to be recognized could be under greater threat.

Areas outside protected areas play important roles in maintaining biodiversity and producing food and livelihoods [4, 11]. They often harbor cultural heritage as well. The conceptualization of and research on traditional land use patterns and associated practices as contribution to conservation and sustainable use of biodiversity originated in Japan, where such landscapes are referred to as Satoyama [12–15]. Landscapes of similar characteristics and function, however, exist elsewhere in the world as well [16]. The Satoyama Initiative is a global effort to realize society in harmony with nature by promoting and supporting such landscapes [17], and it has been recognized by the Convention on Biological Diversity (CBD; Decisions X/32 and XI/25). Under the Satoyama Initiative, the term socio-ecological production landscapes and seascapes (SEPLS) is used to refer to "dynamic mosaic of habitats and land and sea uses where the harmonious interaction between people and nature maintains biodiversity while providing humans with the goods and services needed for their livelihoods, survival and well-being in a sustainable manner" [17; p.3]. These SEPLS have deep links to local culture and knowledge [17]. Landscapes with these characteristics are used for production activities while simultaneously maintaining biodiversity [18]. SEPLS are similar in concept to the cultural landscapes in Europe [19].

The broad definition of SEPLS has the benefit of being inclusive, as attested by the growing membership of the International Partnership for the Satoyama Initiative (IPSI), a global network of 267 organizations (as of June 2020), including international organizations, national and local governments, non-governmental organizations, research institutions, and the private sector. The IPSI database holds nearly 200 case studies on SEPLS (as of May 2020, see https://satoyama-initiative.org/case_study/) contributed by its members from relevant scholarly work from all around the world.

The world agreed to expand the quantity and quality of areas protected by effective systems of protected areas and other effective area-based conservation measures (OECMs) with Aichi Biodiversity Target 11. The CBD defines the OECM as "a geographically defined area other than a Protected Area, which is governed and managed in ways that achieve positive and sustained long-term outcomes for the in-situ conservation of biodiversity, with associated ecosystem functions and services and where applicable, cultural, spiritual, socio-economic, and other locally relevant values" ([20], Paragraph 2). Many SEPLS meet this definition.

Numerous studies on global conservation prioritization have been conducted from diverse perspectives, including the economic efficiency of area-based conservation [4], human impacts and modifications [21, 22], nature of response (proactive/reactive and vulnerability/irreplaceability) [23], and global climate and biodiversity commitments and their synergies [24–26]. Mapping cumulative human modifications using 13 stressors including agriculture, Kennedy et al. [22] argued that moderately modified regions require particular conservation attention and that further prioritization of these regions is needed to capture areas of conservation importance and balance conservation and development. In this regard, SEPLS seem to be prime candidates as a conservation priority. The identification of SEPLS has been bottom up in that sites have been recognized locally. Areas that have been recognized as SEPLS, such as those included in the IPSI database mentioned above, have become conservation priorities [27]. However, there may be many more potential SEPLS around the world that have not been (and probably will not be) recognized in this way. If landscapes and seascapes of diverse qualitative aspects and values (e.g., as discussed in [28], including complex, dynamic, and adaptive systems with biocultural diversity; systems managed through time-tested practices, local

innovations, and decentralized operations; and systems focused on the sense of identity) can be mapped, SEPLS can be mainstreamed into conservation and development policies. The ability to map SEPLS will assist in specifying the areas of importance for biodiversity in production landscapes in the new global biodiversity framework beyond 2020.

Kadoya and Washitani [18] developed the Satoyama Index as a way to map terrestrial SEPLS (or SEPL) globally. It is based on the diversity of land classes within an agricultural landscape as obtained from land cover/use maps. The Satoyama Index values are significantly correlated with the occurrence of the grey-faced buzzard (Butastur indicus) and the richness of amphibian and damselfly species in Japan [18]. Imai et al. [29] determined the effect of spatial resolution and extent of the unit of analysis of the Satoyama Index that was best suited for the landscapes of Japan, and Yoshioka et al. [30] used the Satoyama Index to classify landscapes for land use policy recommendation. Yoshioka et al. [31] further improved the Satoyama Index to account for the degree of dissimilarity between land use types. Although Kadoya and Washitani [18] showed that the Satoyama Index captures areas of importance for biodiversity in ancient agropastoral farming systems in the Iberian Peninsula and shade-grown coffee landscapes in El Salvador, the index has yet to be examined at the global scale. Additionally, although the Satoyama Index focuses on biodiversity and has been validated for that perspective, it has not been tested for its effectiveness in identifying priority sites from the wide spectrum of interests represented in SEPL. The Cultural Landscape Index is a recent development that characterizes rural landscapes, but it uses datasets specific to Europe [19]. The design of the Satoyama Index is more generally applicable globally. In this study, we determine whether the Satoyama Index is effective for identifying SEPL in multi-use, cultural landscapes globally, such as those valued under the Satoyama Initiative.

## Materials and methods

### Land use and land cover map

For the land use and land cover map, we used the most recent dataset of the Global Land Cover by National Mapping Organizations (GLCNMO) provided by the Secretariat of the International Steering Committee for Global Mapping and developed from MODIS imagery taken in 2013 (GLCNMO2013, available at https://globalmaps.github.io/glcnmo.html) [32]. This year of production coincides with the timing of the production of other datasets used in this study. GLCNMO2013 has 20 land cover classes per the Land Cover Classification System of the Food and Agriculture Organization (FAO) [33] (Table 1), and it covers the entire planet

**Table 1. Land cover classifications used for this study.**

| Code | Class Name | Code | Class Name |
|------|-----------|------|-----------|
| 1 | Broadleaf Evergreen Forest | 11 | Cropland |
| 2 | Broadleaf Deciduous Forest | 12 | Paddy field |
| 3 | Needleleaf Evergreen Forest | 13 | Cropland/Other Vegetation Mosaic |
| 4 | Needleleaf Deciduous Forest | 14 | Mangrove |
| 5 | Mixed Forest | 15 | Wetland |
| 6 | Tree Open | 16 | Bare Area, Consolidated (Gravel, Rock) |
| 7 | Shrub | 17 | Bare Area, Unconsolidated (Sand) |
| 8 | Herbaceous | 18 | Urban |
| 9 | Herbaceous with Sparse Tree/Shrub | 19 | Snow/Ice |
| 10 | Sparse vegetation | 20 | Water Bodies |

Source: GLCNMO based on the Land Cover Classification System of the FAO.

in 15-second or approximately 500-m grids (or pixels) provided in the GeoTiff format. The unit of analysis is defined as a 180-second (or approximately 6-km) grid, and we refer to a 6-km cell as a land unit. In the development of the Satoyama Index, Kadoya and Washitani [18] used the GLCNMO2003, which used the same land cover classification but in 30-second or approximately 1-km grids. Although the temporal comparisons of the Satoyama Index values would provide an additional layer of useful information, particularly on the speed of changes, we did not compare the Satoyama Index values between 2003 and 2013 to avoid confusion from false differences due to differences in spatial scales.

We compared GLCNMO2008 and GLCNMO2013, which were produced in the same spatial scale, to see if we could obtain insights from temporal comparison. Since the comparison gave us reasons to suspect that changes due to satellite imagery classifications overwhelmed actual changes in land cover, we did not incorporate 2008–2013 temporal differences in our analysis.

We overlaid a 180-second or 6-km grid on GLCNMO2013 and selected land units that contained at least one pixel of agricultural land cover (Codes 11, 12, or 13 in Table 1). For each land unit (approximately 6 km × 6 km), we determined the number of pixels (approximately 500 m × 500 m) in each land cover class occurring within the land unit using the ArcGIS10.6 Spatial Analyst Tool, Zonal/Zonal Statistics as Table.

## Satoyama Index calculation

Kadoya and Washitani [18] defined the Satoyama Index (*SI*) as the product of the Simpson Diversity Index (*SDI*) and the proportion of natural elements *P* (i.e., *SI* = *SDI* × *P*) per land unit, as follows:

$$SDI = 1 - \sum_{i=1}^{S} p_i^2 = 1 - \frac{n_1^2 + n_2^2 + \cdots + n_{20}^2}{N^2},$$

where *i* is the land cover class, $p_i$ is the proportion of the land unit occupied by land cover class *i*, *S* is the total number of land cover classes (i.e., 20), $n_i$ is the number of pixels of land cover class *i* in a land unit, and *N* is the total number of pixels in a land unit (i.e., 144); and

$$P = \sum_{natural} \frac{n_i}{N - 1},$$

where the summation is for all land cover classes other than urban (Code 18) or agriculture (i.e., cropland [Code 11] and paddy field [Code 12]). The denominator *N*– 1 was used to make *P* range from 0 to 1, since at least 1 pixel was considered agriculture due to how the land units were selected for the Satoyama Index computation.

## Satoyama Index validation

To determine the effectiveness of the Satoyama Index in identifying SEPL globally, we compared its values for known SEPL sites to those of random sites. The datasets and methods are described below, and relevant geographic data in the Shapefile format are provided in the S1 File.

## GEF-Satoyama Project sites

Two datasets were used for representing the SEPL sites (Fig 1). The first set contained the boundaries of the project sites that were digitized from the maps contained in the proposals submitted in response to the calls for proposals for funding by the Global Environment Facility

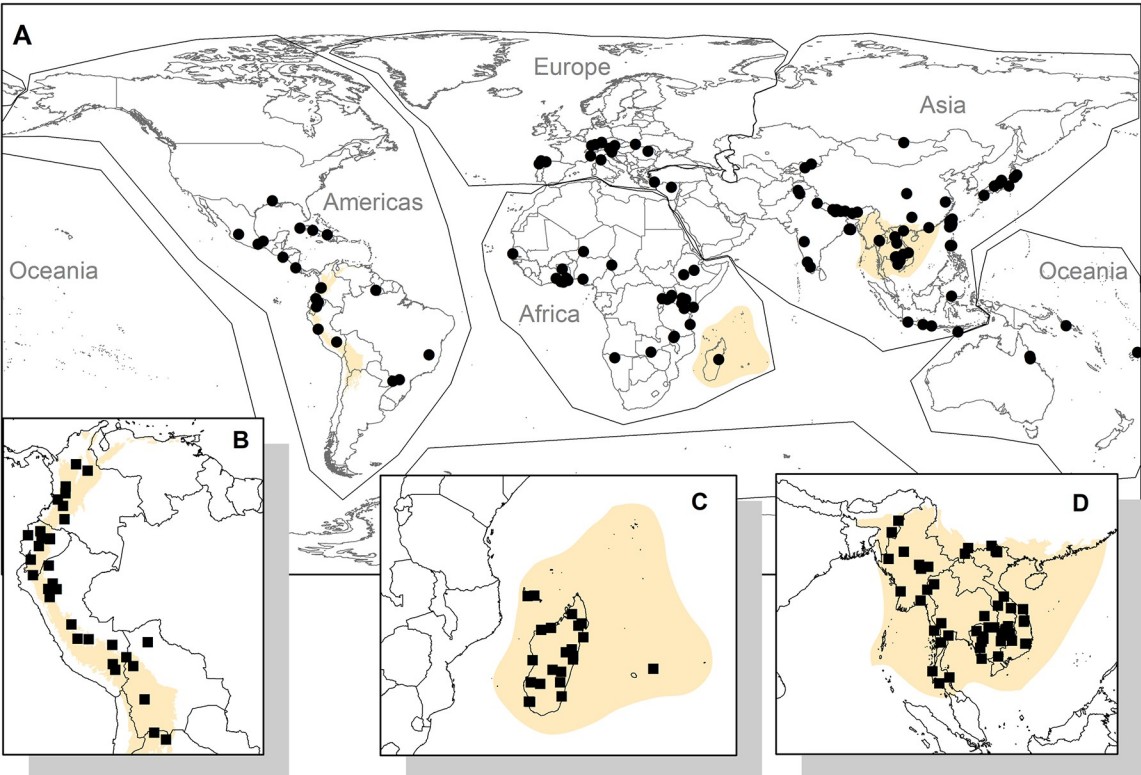

**Fig 1. Locations of SEPL sites.** The map shows the approximate centers of the (A) IPSI case study sites as well as the centroids of polygons of the GEF-Satoyama Project sites in the biodiversity hotspots of (B) the Tropical Andes, (C) Madagascar and the Indian Ocean Islands, and (D) Indo-Burma. The map also shows the boundaries of the regions used for this study. (Made with Natural Earth; free map data at naturalearthdata.com.).

(GEF) under the "GEF-Satoyama Project" (http://www.thegef.org/projects; Project ID: 5784). The calls for proposals targeted three biodiversity hotspots [34]: Indo-Burma, the Tropical Andes, and Madagascar and the Indian Ocean Islands. The solicited projects had to focus primarily on mainstreaming conservation and the sustainable use of biodiversity and ecosystem services resulting in improved human wellbeing through a) conserving, maintaining, or revitalizing traditional sustainable practices, threatened species, and/or sites of global significance for biodiversity conservation; b) restoring degraded production landscapes and/or seascapes; and/or c) implementing livelihood alternatives (e.g., sustainable agricultural, fisheries, or forestry production techniques for the sustainable use of terrestrial, freshwater, or marine systems or a combination of these). A total of 109 terrestrial sites were digitized.

## IPSI case studies

The second dataset used for the SEPL sites contained the locations of landscapes covered in case studies that were submitted to the IPSI by its members (available at http://satoyama-initiative.org). The steering committee of the IPSI evaluates membership applications for relevance of the organization's activities and experiences to the objective and focus of the IPSI. The submission of at least one case study is the requirement of each admitted member. As many case studies lacked maps for the delineation of exact boundaries, we located the most likely centers of the case study sites from maps or verbal descriptions provided in the case studies. We identified the 3 × 3 clusters of land units (i.e., nine 6-km grid cells), such that the

central land units of the clusters contained the center of the case study sites (referred to as nine neighbors). Many case study sites were larger than these nine neighbors, resulting in under-representation in terms of spatial coverage. There were, however, a few sites where the nine neighbors covered areas beyond the case study boundaries. One hundred twenty-six nine neighbors had at least one Satoyama Index value determined.

### Random sites

Because of the differences in how sites are identified and expressed in the two datasets above, we treated them separately in the statistical analysis. For the purpose of statistical analysis, we generated two sets of random nine neighbors. One set was drawn randomly from terrestrial areas between latitudes of 75˚ North and 60˚ South to be used with the IPSI case study sites. Of 959 random nine neighbors on terrestrial areas, 529 had at least one Satoyama Index value determined. Another set, to be used with the GEF-Satoyama Project sites, was drawn randomly at a higher density for the three biodiversity hotspots targeted by the Project. Of the 153, 169, and 124 random nine neighbors drawn for Indo-Burma, the Tropical Andes, and Madagascar and the Indian Ocean Islands, 143, 134, and 123 of them, respectively, had at least one Satoyama Index value determined.

### Statistical analysis

We tested the null hypothesis that the medians and maxima of the Satoyama Index in SEPL sites do not differ from those of random sites against the alternative hypothesis that they are higher than expected from random (i.e., one-sided) using Wilcoxon's rank sum test in R (version 4.0.0) at $p < 0.05$. We chose a nonparametric test to deal with outliers and non-normal distributions. We tested the GEF-Satoyama Project sites by biodiversity hotspot and IPSI case study sites by groupings of Africa, Americas, Asia, Europe, and Oceania (Fig 1). We compared the median Satoyama Index value of the land units in each site to minimize the influence of outliers. We also compared the maximum Satoyama Index values of the land units in each site in the consideration of our observation that the delineation of the boundaries of projects and case studies often follow existing boundaries from other purposes that contain extensive non-SEPL areas as well; we deemed that the use of maximum Satoyama Index values to represent the area would capture the very features that made the areas SEPL better than the median values would.

### Dataset for areas of biodiversity importance

We used the following GIS files for analysis of the Satoyama Index with respect to areas of biodiversity importance. For recognized (biologically or politically) protected areas, we used the protected areas' polygon data in the World Database on Protected Areas dataset downloaded from the Protected Planet website (https://www.protectedplanet.net/; accessed on March 29, 2019). We considered terrestrial protected areas in all protected area categories that were recorded as "inscribed," "adopted," "designated," or "established" (215,918 polygons). We considered land units to be inside a protected area if their centers were inside protected areas (966,163 land units).

The second dataset was that of KBAs. KBAs are sites of importance for biodiversity in the short term, which are scientifically identified per internationally standardized criteria [35–37]. We obtained the polygon data of the world's KBAs from Birdlife International [38] (15,074 KBAs) and determined the land units inside each KBA in the same manner as we did for protected areas (479,706 land units). These polygons were overlaid with the Satoyama Index map in ArcGIS10.6 to identify the spatial overlaps.

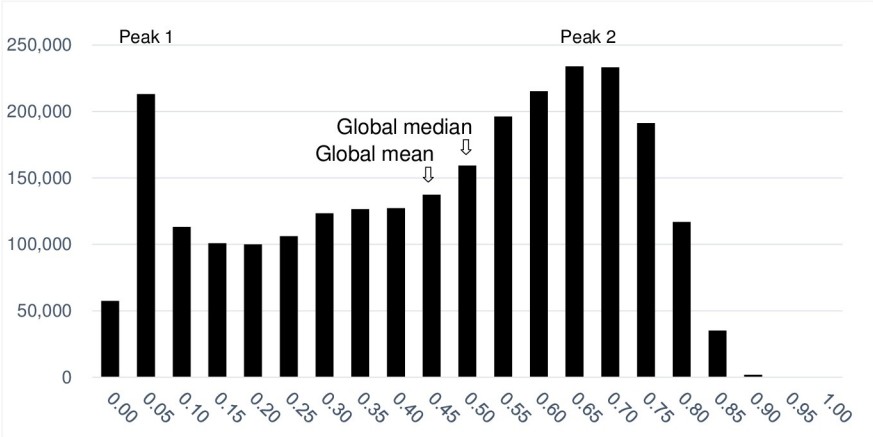

**Fig 2. Histogram of the Satoyama Index values.** The bins are in 0.05 increments and the x-axis labels show the upper bound of the bins (inclusive).

## Results

### Mapping

The Satoyama Index was calculated for 2,588,370 land units out of 8,571,076 terrestrial land units, and the values ranged from 0 to 0.8965 with a mean of 0.4299 and median of 0.4792. Their distributions had two peaks: one around 0.00–0.05 and another around 0.60–0.70 (Fig 2). On a continental scale, there were concentrations of high Satoyama Index values in southeastern China, the Himalayas, the European Alps, Madagascar, eastern Africa to the south of the equator, the Tropical Andes, the southern and eastern Amazon and Cerrado in Brazil, southeastern United States, western Canada, and western Russia and a line across Siberia (Fig 3). In contrast, concentrations of low Satoyama Index values were found in northern and eastern China, India, areas around the Black Sea, southeast and southwest Australia, central United States and Canada, Argentina, and Kazakhstan. The Satoyama Index captured areas of local importance at more local scales; for instance, a concentration of high values on the absolute scale or otherwise considerably higher values compared to the surroundings were found in Western Ghats in India. The results in the Shapefile format are provided in the S2 File.

### Validation

The medians of the Satoyama Index values of the SEPL sites were not significantly greater than the random sets ($p > 0.05$) except for the IPSI case study sites in Asia ($p = 0.0069$; Table 2). However, the maxima of the Satoyama Index values of the SEPL sites were found to be significantly greater than random almost globally ($p < 0.05$; Table 2), with the exception of the Madagascar and Indian Ocean biodiversity hotspot ($p = 0.1528$). The number of IPSI case studies were small in Oceania ($n = 3$), and a discussion on this region should wait until more sites have been studied.

## Discussion

### Global applicability of the Satoyama Index

Datasets covering different regions of the world and different forms of human uses in various ecosystems demonstrated the applicability of the Satoyama Index globally, although it was primarily developed and previously validated only within the local context of Japan [18]. The

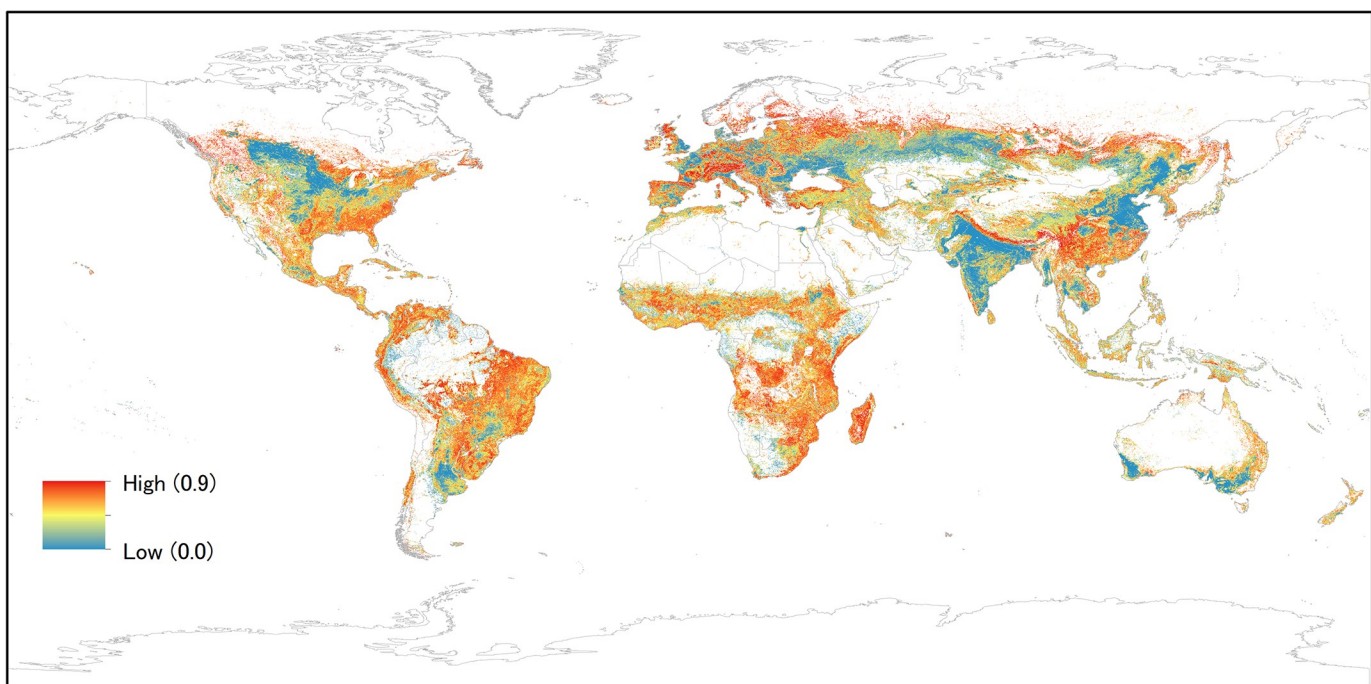

**Fig 3. The global map of the Satoyama Index values.** The Satoyama Index (range: 0–1) was calculated for each 6 km × 6 km land unit with agricultural land use across the globe. Terrestrial areas with no agricultural pixels within the land unit appear in white. (Made with Natural Earth; free map data at naturalearthdata. com.).

design of the Satoyama Index does not include explicit considerations of social and cultural aspects, but the Satoyama Index was found to capture the locations of SEPL on the global scale reasonably well. This enables the geographic presentation of SEPL, which would facilitate mainstreaming of SEPL in conservation policy.

The finding that the medians of the Satoyama Index did not differ from random expectation but the maxima did (Table 2) is consistent with the assumption that the boundaries of projects or studies about SEPL considers broader landscapes than SEPL per se. A project or study might deal with protected area management and work with local communities as a means to achieve this objective. In this case, the project/study site map will contain both the

**Table 2. Comparison of the median and maximum Satoyama Index values between the SEPL sites and randomly generated points by Wilcoxon's rank sum test.**

| Treatment | n | Comparison of medians | | | | Comparison of maxima | | | | Random set compared against |
|---|---|---|---|---|---|---|---|---|---|---|
| | | Treatment mean | Random mean | W | p | Treatment mean | Random mean | W | p | |
| GEF_IB | 50 | 0.472 | 0.440 | 3,928 | 0.1499 | 0.666 | 0.575 | 4,951 | < **0.0001** | IB (n = 143) |
| GEF_TA | 35 | 0.488 | 0.467 | 2,464 | 0.3229 | 0.701 | 0.596 | 3,322 | **0.0001** | TA (n = 134) |
| GEF_MI | 24 | 0.606 | 0.654 | 943 | 0.9974 | 0.754 | 0.746 | 1,672 | 0.1528 | MI (n = 123) |
| IPSI overall | 126 | 0.506 | 0.450 | 37,531 | **0.0138** | 0.675 | 0.571 | 43,108 | < **0.0001** | Global (n = 529) |
| IPSI_Africa | 29 | 0.550 | 0.524 | 1,561 | 0.3563 | 0.692 | 0.624 | 1,797 | **0.0479** | Africa (n = 103) |
| IPSI_Americas | 20 | 0.526 | 0.463 | 2,045 | 0.0839 | 0.701 | 0.588 | 2,412 | **0.0017** | Americas (n = 172) |
| IPSI_Asia | 60 | 0.481 | 0.392 | 6,333 | **0.0069** | 0.660 | 0.519 | 7,095 | < **0.0001** | Asia (n = 174) |
| IPSI_Europe | 14 | 0.530 | 0.439 | 519 | 0.1530 | 0.728 | 0.578 | 603 | **0.0165** | Europe (n = 63) |
| IPSI_Oceania | 3 | 0.322 | 0.510 | 8 | 0.9728 | 0.400 | 0.585 | 9 | 0.9640 | Oceania (n = 17) |

IB: Indo-Burma; TA: Tropical Andes; MI: Madagascar and the Indian Ocean Islands; GEF: GEF-Satoyama Project; IPSI: IPSI case studies.

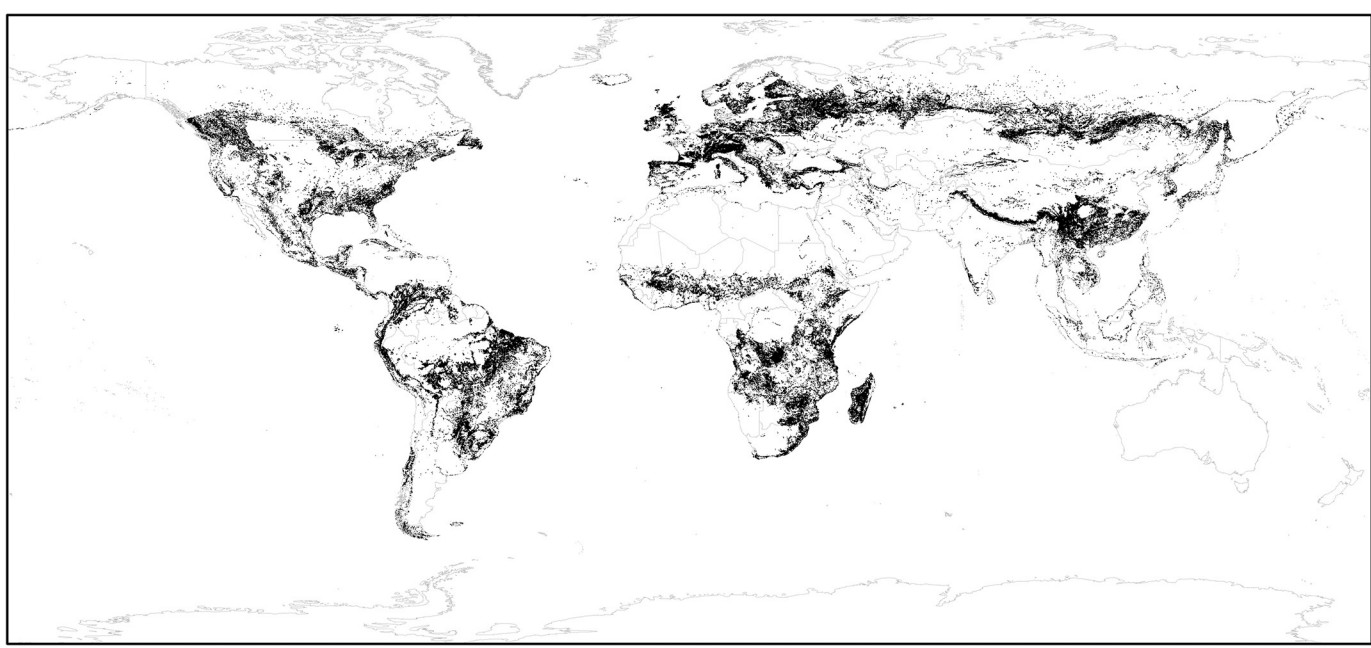

**Fig 4. Potential locations of the SEPL.** The land units with the Satoyama Index values equal to or greater than the threshold of 0.704572 are shown in black. These represent land units around which SEPL may be found. The threshold is not set for Oceania. The edges of land units are exaggerated for better visibility on a global map. (Made with Natural Earth; free map data at naturalearthdata.com.).

location of the community activities and protected area. Protected areas tend to have low or no Satoyama Index values due to the absence of agricultural operations or relatively uniform land covers, as discussed below in the section on areas of biodiversity significance. The maximum Satoyama Index values of both GEF-Satoyama Project sites (represented by the actual project boundaries as documented by the project proponents) and IPSI case study sites (represented by the nine neighbors around the center of the case study sites) supports that the Satoyama Index can capture the location of SEPL as conceived by those involved in the Satoyama Initiative.

One of the important features of SEPL that differentiate them from area designations based on ecological values, which is contained in both GEF-Satoyama Project sites and IPSI case study sites, is the presence of cultural and traditional practices in the landscapes. Although it is not possible to map such practices directly, the finding that SEPL can be identified in areas where the Satoyama Index is high provides a significant step forward for mapping SEPL globally.

The finding that SEPL likely contain land units with higher Satoyama Index values than randomly expected enables us to set Satoyama Index thresholds to identify the nucleus land units around which SEPL boundaries may be defined in practice. Here, we adopted the 50th percentile of the IPSI case study sites other than Oceania as the SEPL thresholds (i.e., 0.704572) (Fig 4). Oceania was excluded because the statistical analysis did not show the effectiveness of the Satoyama Index in identifying SEPL there. We used the values from the IPSI case study dataset because of its broader and more global coverage than the GEF-Satoyama Project dataset. The precise threshold value will vary depending on a number of factors, including data from more SEPL sites for reference points and base land use/cover data used to compute the Satoyama Index, but the map shows the locations where SEPL are likely to be found.

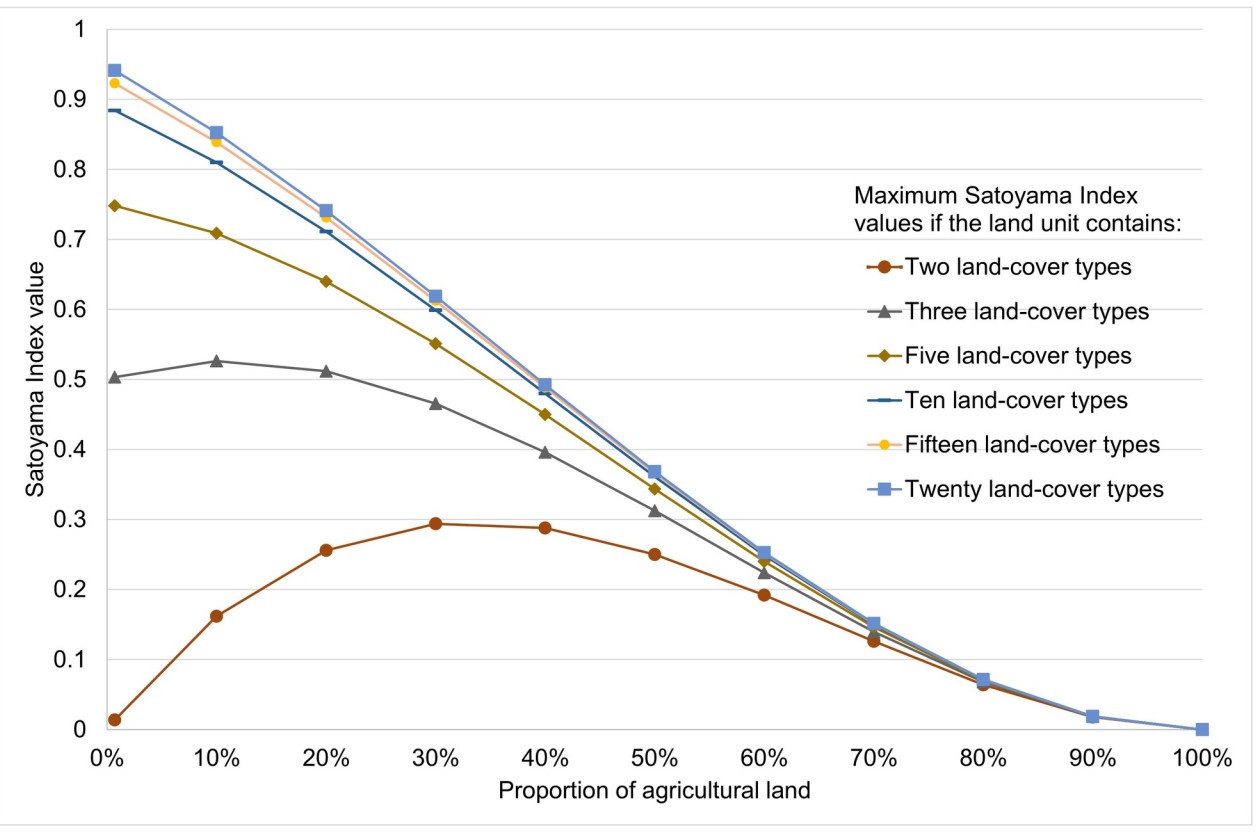

**Fig 5. Theoretical maximum values of the Satoyama Index by number of land cover types in the land unit and by proportion of land unit covered by agricultural land.** Each line represents the theoretical maximum, given the proportion of agricultural land, that is achieved when all land cover types except for agricultural land are distributed in equal proportions. The minimum values for any number of land cover types in the land unit approaches the line for two land cover types. For simplicity, "urban" was not discounted here. If it had been, the curves of the maximum values would have been lower.

For a given land unit, the percentage of cropland and the number of land cover types determined the possible maximum value of the Satoyama Index (Fig 5). The threshold bound of 0.70, as discussed in the previous paragraph, can only be achieved in land units with a low proportion of cropland ($\leq 0.2$) and at least five cover types at more or less equal proportions. Traditional land use of shifting cultivation, which is estimated to cover 280 million ha globally [39], would produce a landscape like this.

## Regions of high Satoyama Index values

A qualitative ocular assessment indicated that there is a high correlation between the complexity of the topography (we used GRAY_HR_SR_W raster data from Natural Earth for this purpose; available at https://www.naturalearthdata.com/) and the Satoyama Index values. This is reasonable because the more complex the terrain is, the more diversity in land use or land cover the area is likely to have. This explains the high Satoyama Index values in areas such as Nepal, southern China, the European Alps, and the Tropical Andes. Further quantitative analyses on the relationship between the Satoyama Index and topography would help provide contextual information about the areas with similar Satoyama Index values.

Another region with high Satoyama Index values is the Amazon in Brazil, where the areas with high Satoyama Index values coincided with deforestation (Fig 6). The land units in which

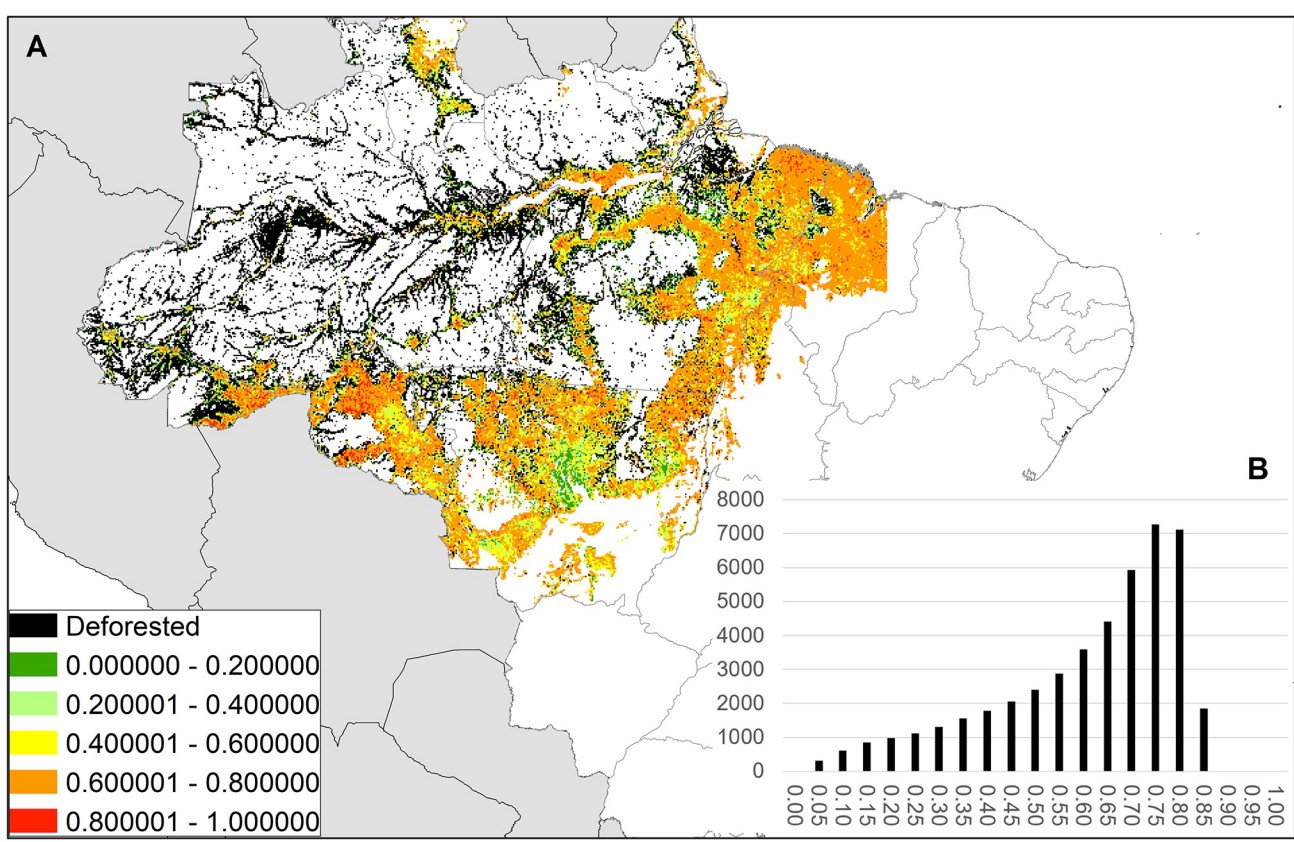

**Fig 6. Satoyama Index values in deforestation areas in the Amazon region of Brazil.** (A) Pixels with color show the values of the Satoyama Index overlapping with deforestation as of 2013. Black pixels are the areas where deforestation had occurred by 2013 but which the Satoyama Index did not compute (i.e., there is not agricultural land cover recorded). (B) Histogram of the Satoyama Index values shown in (A). (Made with Natural Earth; free map data at naturalearthdata.com.).

deforestation have occurred had Satoyama Index values concentrated at its high end (around 0.65–0.80). The Satoyama Index value grew higher as the forest was converted into some other "natural" land cover, such as grassland. We considered this behavior of the Satoyama Index to be highly problematic and discuss the cause below.

We identified the land units containing cumulative deforestation patches through 2013 in the PRODES dataset ([40] and http://www.obt.inpe.br/OBT/assuntos/programas/amazonia/prodes; downloaded from http://www.dpi.inpe.br/prodesdigital/dadosn/mosaicos/2013/). The dominant land cover types in land units containing deforestation patches, as observed in the GLCMNO dataset, were broadleaf evergreen or deciduous forest (i.e., remnant forest patches; 44%), herbaceous or herbaceous with sparse tree/shrub (24%), tree open (13%), and cropland or cropland with other vegetation mosaic (12%). Given that deforestation in the Amazon is driven by commodity production [9], this raises the possibility of systematic confusion in the GLCMNO dataset. An ocular inspection on Google Earth revealed that the relevant areas largely consist of a collection of rectangular plots of different cover types (grassland or cropland with or without sparse tree covers) with rectangular remnants of tall broadleaved forest patches. Because of the mosaic of different land covers this process causes, the deforested areas had high Satoyama Index values. The GLCNMO2013 dataset has a moderate user's accuracy of 57–84% for three cropland types separately (Codes 11, 12, and 13) and high users' accuracy of 88.1% for aggregated cropland [32]. However, since cropland monitoring is found to be

subject to high variation between satellite sensors, classification methods, and country [41], it is possible that the Satoyama Index is overrated in the region of newly deforested parts of the Amazon because of the misclassification of cropland for natural cover types, such as herbaceous or herbaceous with woody vegetation. Indeed, there is a large discrepancy between the GLCNMO and other agriculture statistics. The proportion of cropland in Brazil declined rapidly from 32% in 2003 to 22% in 2008 and 19% in 2013 according to the GLCNMO [32, 42, 43], while FAOSTAT recorded a steady proportion of agricultural land (27–28%; "cropland" + "land under permanent meadow and pastures") during the same period (downloaded from http://www.fao.org/faostat/en/#data/EL on July 24, 2020). Since the agricultural land is not decreasing, the GLCNMO data is likely classifying some cropland as grassland, which results in increasing the SI values substantially. Kadoya and Washitani [18] used the GLCNMO 2003 dataset in their development of the Satoyama Index and showed low index values across the Amazon.

## Regions of low Satoyama Index values

The lack of land cover diversity and dominance of anthropogenic land cover (e.g., agriculture and urban) led to low Satoyama Index values. Vast expanses of cropland had low Satoyama Index values in India, eastern China, the area around the Black Sea, southeast and southwest Australia, central United States and Canada, Argentina, and Kazakhstan. Cropland in these regions are qualitatively different. Landholdings in India and China are predominantly small (5 ha or smaller), while they are large (50 ha or larger) i n Argentina [44, 45]. A land unit of 6 km × 6 km could be too large in such a context. Since SEPL do not exist in isolation (i.e., one land cover type alone does not make an area an SEPL, but the combination of several land cover types can), using the broad land unit as the unit of analysis for the Satoyama Index is an appropriate approach. However, we expect there to be more qualitative differences arising from the difference in the sizes of landholdings. As agriculture is intensified, landscapes get simplified and lose biodiversity [46]. Agricultural intensification is more likely associated with large landholders than small holders. If this holds true, small holder farms are likely to have more heterogeneity within the farm and across farms; thus, they will have more fine-scale diversity as a landscape. Besides these ecological aspects, there could be different cultural and social characteristics unique to different regions, which are an important part of SEPL. Further investigation into the contribution of small holders in maintaining biodiversity in production landscapes is a timely subject now, as 2019–2028 is the United Nations Decade of Family Farming [47].

## Points for improvement

Since the Satoyama Index uses cross-section data, it does not reflect histories of land uses and land-use changes. Heterogeneity in landscape present different significance, whether it is in steady states, which the Satoyama Initiative inherently assumes, or in transient states, such as one created by deforestation and farmland abandonment [48]. The concerns about land-use changes that may counter biodiversity conservation appearing high in the Satoyama Index, as discussed with the case of the Amazon, may be addressed through a use of mask layer that apply discount coefficients for such changes mapped by comparison of satellite imageries at two different times. Such a mask layer will need to consider the context-dependent nature of the impact of land-use changes on biodiversity [48]. Similarly, another masking layer can be used to incorporate the variation in land-holding sizes that we discussed above.

Since the Satoyama Index is designed for assessments at global scale, it is not sufficiently sensitive to more locally focused interests. Landscape heterogeneity may contribute to

biodiversity of agricultural landscapes [49–52]. The size of landholdings may serve as a surrogate to farm sizes that relate to ecological functions. Regions such as India, China, Sub-Saharan Africa, and Southeast Asia tend to have small landholdings [44, 45], and thus they may have higher conservation values that are not captured in the current Satoyama Index. Fine-scale land use practices, such as soil ditches vs. concrete ditches around rice paddies [53], can produce large differences in habitat conditions for those species closely associated with agricultural landscapes, which is not detectible at the scale of the current Satoyama Index, either. A study in rural Japan found a finer resolution of input data (50 m), while maintaining the same extent of the land unit (6 km), best explained the local biodiversity [29]. In Addition, the incorporation of contrasts between similar and dissimilar land cover types is one method for considering features of more local interest, which can improve the ability of the index to explain biodiversity [31]. Similar studies may be replicated elsewhere under different land use systems (such as landscapes of grazing land and dry farms) to capture the characteristics of those specific landscapes better. Since data sizes, quality, and availability prevent the collection of such fine spatial resolution data for global coverage, more detailed studies should be conducted at more local scales in places with high Satoyama Index values to locate the quality SEPL more precisely and in places with low Satoyama Index values but with known SEPL quality to identify additional factors that can inform future improvement of the index.

Many studies have found that landscape configuration heterogeneity contributes more strongly to biodiversity than compositional diversity [49–52]. However, crop diversity (a compositional diversity) is important for SEPL as the physically observable surrogate of cultural practices [54, 55]. Cultivation of diverse crop varieties, such as potatoes in the Andes [54] and corn in Mexico [55], is a traditional practice, which is among the defining features of SEPL. Shifting cultivation also represents traditional cultural practices [56], but it, too, is often not captured in land cover maps generated from satellite imagery [39]. Due to the thematic resolution of global land cover maps (GLCNMO for this study, but other datasets as well), the Satoyama Index cannot explicitly address this aspect. To address this issue, we suggest that the diversity of cultivated crops should be dealt with at finer scales (resolution of tens of meters and extent of hectares to hundreds of hectares) in the areas that the Satoyama Index identifies the potential to be SEPL broadly.

### Areas of biodiversity significance

The overlay of the Satoyama Index with protected areas revealed that a large majority (79.2%) of land units with the Satoyama Index values equal to or greater than the SEPL threshold was outside protected areas or KBAs (Fig 7). This means that SEPL are not recognized as important for conservation under a conventional sense. Biodiversity conservation is dependent on human-influenced landscapes [57], but identification of conservation priority sites in production landscapes has been inadequate to date. This deficiency may be complemented by OECMs. OECMs are areas effectively managed outside protected areas that contribute to insitu conservation of biodiversity. The identification of OECMs should recognize SEPL around the world, as this recognition would support the conservation of many SEPL.

### Conclusions

We provided a method for addressing the lack of information on the geographic distribution of SEPL by enabling their mapping globally. We demonstrated that the Satoyama Index, which uses purely physical variables, can capture sites of social and cultural aspects as well. We also demonstrated that the global applicability of the Satoyama Index in identifying SEPL, which the original study that introduced the Satoyama Index lacked [18].

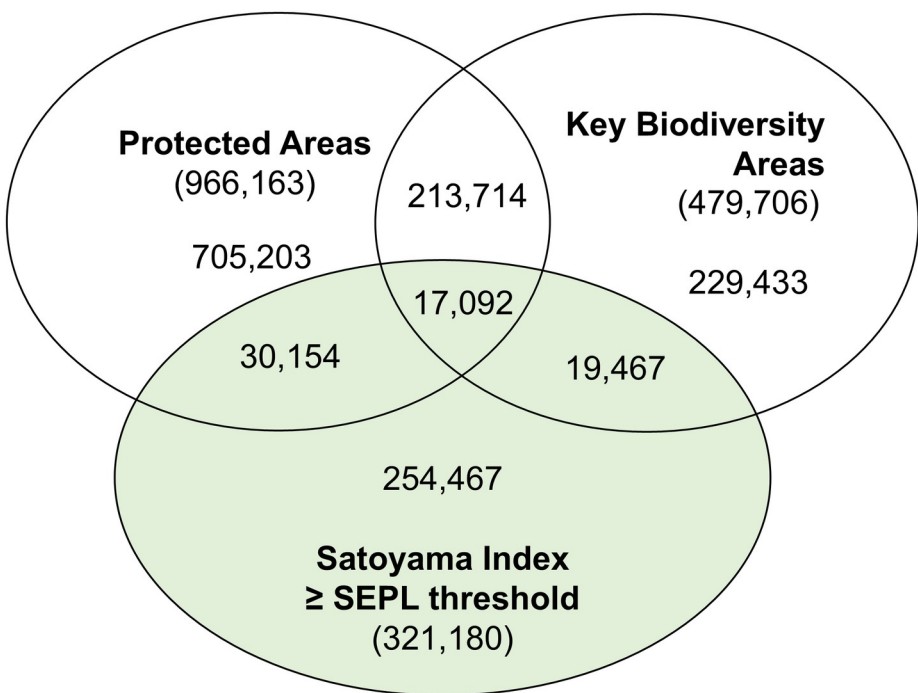

**Fig 7. The number of land units by conservation categories.** The numbers in parentheses represent the number of land units in the category.

SEPL hold the potential to harmonize conservation and livelihood, but 80% of them are found to be outside conservation priority sites recognized as protected areas and KBAs. Recognizing SEPL as OECMs is one of the ways to mobilize public and private interests towards their appropriate management. Biodiversity conservation is a goal that should be pursued in all lands and optimized for the given circumstances, as recommended by the Aichi Biodiversity Target 11 (area-based conservation of biodiversity) and the Sustainable Development Goal 15 (sustainable management of the terrestrial environment). By enabling easy comparison of the sites geographically, mapping can help identify synergies with initiatives that are unaware of the Satoyama Initiative or its international partnership (IPSI) or those that engage in similar activities under different names. This can help decision makers to identify potential trade-offs that can occur on the same piece of land, such as expanding and intensifying agriculture to address hunger issues vs. terrestrial conservation.

The inclusion of histories of land-use changes and sizes of agricultural landholdings can improve the SEPL identification with the Satoyama Index further on the global scale. Refinement and fine tuning of spatial scales and thematic resolution need to be considered for more local application, such as prioritization of area-based conservation in national or subnational policies.

## Supporting information

**S1 File. GIS files created for the analyses in this article.**
(RAR)

**S2 File. The Satoyama Index values in the shapefile format.**
(RAR)

## Acknowledgments

This study benefited from discussions with Dr. Taku Kadoya of the National Institute for Environmental Studies; Dr. Federico Lopez-Casero, Mr. Yasuo Takahashi, and Dr. Ikuko Matsumoto of the Institute for Global Environmental Strategies; Mr. Yohsuke Amano of the United Nations University Institute for the Advanced Study of Sustainability; Dr. Devon Dublin of Conservation International Japan; and the members of the International Partnership for the Satoyama Initiative. (Affiliations listed were at the time of research).

## Author Contributions

**Conceptualization:** Yoji Natori.

**Data curation:** Yoji Natori, Akihiko Hino.

**Formal analysis:** Yoji Natori, Akihiko Hino.

**Funding acquisition:** Yoji Natori.

**Methodology:** Yoji Natori.

**Supervision:** Yoji Natori.

**Visualization:** Yoji Natori.

**Writing – original draft:** Yoji Natori.

**Writing – review & editing:** Akihiko Hino.

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
