## [Decision Letter · Decision Letter 0]

18 Mar 2021

PONE-D-20-26225

Global identification and mapping of socio-ecological production landscapes with the Satoyama Index

PLOS ONE

Dear Dr. Natori,

Thank you for submitting your manuscript to PLOS ONE. After careful consideration, we feel that it has merit but does not fully meet PLOS ONE’s publication criteria as it currently stands. Therefore, we invite you to submit a revised version of the manuscript that addresses the points raised during the review process.

Thank you for your manuscript. We acquired reviews from four reviewers and they vary widely, ranging from a recommendation to reject (Reviewer 2) to major and minor revision recommendations (Reviewers 1 and 3) and even an "accept with editing for English usage" (Reviewer 4). After reading through the reviews and also the manuscript, I believe some somewhat substantial revisions are necessary to improve the manuscript and make it appropriate for publication. Specifically, please address the comments of Reviewer 1, Reviewer 3, and Reviewer 4, which are either fairly detailed (as in recommending specific edits and changes) or a bit general (improve English usage). Each of these reviewers indicates some clear changes, but also asks some specific questions. Please be sure to make changes which respond to these.

Although I did not chose to fully accept reviewer 2's recommendation to reject this manuscript at this time, I do believe you must add a section addressing how the Satoyama Index can be extended to apply to more broadly defined landscapes, or why Reviewer 2's assessment has shortcomings. In this section, please address a broad audience, not just Reviewer 2, but be sure to do justice to their statements.

Finally, I agree with Reviewer 1 that the data should be made available in an open format if possible, but would note that a File Geodatabase (*.gdb) is accessible in open source software through the GDAL framework (https://gdal.org/drivers/vector/openfilegdb.html). If possible please convert, but if not, I would note that in your response you should indicate how these data can be accessed in open source GIS software.

We look forward to receiving your revised manuscript.

Kind regards,

Stephen P. Aldrich, PhD

Academic Editor

PLOS ONE

Journal Requirements:

'This study was funded in part by a grant from the Global Environment Facility to Conservation International Japan for the “GEF-Satoyama Project” (Project ID: 5784; http://www.thegef.org/projects). The funders had no role in study design, data collection and analysis, decision to publish, or preparation of the manuscript.'

a. Please provide an amended statement that declares *all* the funding or sources of support (whether external or internal to your organization) received during this study, as detailed online in our guide for authors at http://journals.plos.org/plosone/s/submit-now

Please also include the statement “There was no additional external funding received for this study.” in your updated Funding Statement.

4. We note that Figures 1, 3, 4 and 6 in your submission contain map images which may be copyrighted.

4a.          You may seek permission from the original copyright holder of Figures 1, 3, 4 and 6 to publish the content specifically under the CC BY 4.0 license. 

4b. If you are unable to obtain permission from the original copyright holder to publish these figures under the CC BY 4.0 license or if the copyright holder’s requirements are incompatible with the CC BY 4.0 license, please either i) remove the figure or ii) supply a replacement figure that complies with the CC BY 4.0 license. Please check copyright information on all replacement figures and update the figure caption with source information. If applicable, please specify in the figure caption text when a figure is similar but not identical to the original image and is therefore for illustrative purposes only.

5. Please ensure that you refer to Figure 5 in your text as, if accepted, production will need this reference to link the reader to the figure.

Additional Editor Comments:

Thank you for your manuscript. We acquired reviews from four reviewers and they vary widely, ranging from a recommendation to reject (Reviewer 2) to major and minor revision recommendations (Reviewers 1 and 3) and even an "accept with editing for English usage" (Reviewer 4). After reading through the reviews and also the manuscript, I believe some somewhat substantial revisions are necessary to improve the manuscript and make it appropriate for publication. Specifically, please address the comments of Reviewer 1, Reviewer 3, and Reviewer 4, which are either fairly detailed (as in recommending specific edits and changes) or a bit general (improve English usage). Each of these reviewers indicates some clear changes, but also asks some specific questions. Please be sure to make changes which respond to these.

Although I did not chose to fully accept reviewer 2's recommendation to reject this manuscript at this time, I do believe you must add a section addressing how the Satoyama Index can be extended to apply to more broadly defined landscapes, or why Reviewer 2's assessment has shortcomings. In this section, please address a broad audience, not just Reviewer 2, but be sure to do justice to their statements.

Finally, I agree with Reviewer 1 that the data should be made available in an open format if possible, but would note that a File Geodatabase (*.gdb) is accessible in open source software through the GDAL framework (https://gdal.org/drivers/vector/openfilegdb.html)

Reviewers' comments:

Reviewer's Responses to Questions

**Comments to the Author**

1. Is the manuscript technically sound, and do the data support the conclusions?

Reviewer #1: Yes

Reviewer #2: No

Reviewer #3: Yes

Reviewer #4: Yes

2. Has the statistical analysis been performed appropriately and rigorously? 

Reviewer #1: Yes

Reviewer #2: Yes

Reviewer #3: Yes

Reviewer #4: Yes

3. Have the authors made all data underlying the findings in their manuscript fully available?

Reviewer #1: No

Reviewer #2: Yes

Reviewer #3: Yes

Reviewer #4: Yes

4. Is the manuscript presented in an intelligible fashion and written in standard English?

Reviewer #1: Yes

Reviewer #2: Yes

Reviewer #3: Yes

Reviewer #4: No

5. Review Comments to the Author

Reviewer #1: This manuscript reported that socio-ecological production landscapes (SEPLS) can be mapped using the Satoyama Index (Kadoya & Washitani 2011), an indicator of traditional agricultural landscapes with high biodiversity. Following the reviewer’s comments, the authors substantially revised the manuscript; they vigorously collected data on spatial distribution of SEPLs (terrestrial socio-ecological production landscapes) and conducted statistical test to compare the values of Satoyama index inside and outside the SEPLs. Although the manuscript was greatly improved, some modifications will make it more suitable for publication. Please see comments below.

L51-79. Rearrangement of the paragraphs will make the part easier to read. For example, the part on “SEPLS and area-based conservation prioritization” (L51-59) can be moved to immediately after the part on OECM (L70-79). Then, the sentences in L66-69 can be moved to head of the paragraph on importance of geographic information of SEPLS(L80-84). Please note that some corrections of sentences should follow the rearrangement (For example, “At the same time, it…” should be corrected to “However, IPSI database presented a weakness that…” if the sentences moved to the head of the L80.)

L64-66. Although the authors assumed that there are potential SEPLs which have not registered in the IPSI database (and will be found by Satoyama index) in the world, it was not explicitly mentioned here. The large number of the cases in the IPSI database can give readers the impression that the database covers the SEPLs in the world. So, I recommend that authors explicitly mention the assumption here.

L189. Please clarify the spatial scale of a 3*3 clusters (6km * 6km?). I think that the authors meant 3*3 cells (or pixels) here, it should be difficult to understand for readers unfamiliar with GIS data.

L190. Please correct “he” to “the”.

L193-194. Is the dataset of “9-neighbors” included in the Supplement material? If so, please cite it (e.g. data S1 etc.).

L211. Was the assumption of normality of the data checked? (Although I do not think it critically affect the result)

In addition, I recommend the authors to attach that the raw data used in the statistical analysis in the form of an Excel file (or csv or txt) as a supplement.

L279. Fig. 7 should be cited here.

L305. Citation error. Fig. 5?

L319-361. The problem on Amazon does not appear to be improved by the point mentioned in the “points for improvement”. Considering that newer land classification had made the problem more serious, simple improvement of satellites or base landuse/landcover maps may not be effective. So, please add the idea (or hypothesis) which may systematically resolve the problem to the end of the section or the part on the points for improvement. For example, consideration of the history of farming may be a key. The result of meta-analysis by Queiroz et al. (2014) suggested that the contribution of land abandonment to biodiversity differed between the old and new continents. In addition, consideration of spatial unit of unsustainable crop rotation may also be important. In the Amazon, newer GLCNMO may correctly classified wild (abandoned?) open habitats owning to crop rotation, which is usually unsustainable and less important for biodiversity in the region. The problem of overvaluation of non-crop habitats following crop rotation may have been overlooked in the region where rice paddies are dominant, because rice paddies do not need crop rotation from the view point of sustainability. So, there is a room for improvement in the Satoyama index, which originally focused agricultural landscapes including rice paddies.

Queiroz et al. 2014

https://esajournals.onlinelibrary.wiley.com/doi/abs/10.1890/120348

L670. The “gdb” is a special file format for ArcGIS. The file should be shared in the form which can be opened by the other free software such as QGIS.

Reviewer #2: The manuscript presents a very simple index developed for specific agricultural and sustainable-use landscapes in Japan that are often protected. The index can work very well in such landscapes where the heterogeneity of the landscape associated with the cultural model is constant. But I don't see that it can work in other types of landscapes with different degrees of heterogeneity associated with different cultural models. In fact the authors find abnormally high and low values in different circumstances. The applicability of this simple index, which does not consider the social economic model, nor the associated ecological functioning of landscapes, seems to me to be impossible for the detection of productive socio-ecological landscapes in places other than the productive landscapes of Japan. It can work to detect differences between the productive landscapes of Japan, but the idea of its global applicability does not seem possible to me. However, the manuscript is well written and statistically adequate. I would recommend the authors to adapt the index for use in a globalized framework.

Reviewer #3: Major comments:

The paper outlines the use of the Satoyama Index and its usage and potential for mapping and understanding diverse production landscapes. This reviewer is not an expert in statistical analysis, but it appears to make a good case for its argument, based on the research and analysis provided. One point is that it could well use a native proofreader to look through it as there are many minor issues with the language.

Minor comments (I will not note the many minor errors in English that could be addressed by a native check but rather focus on substance and only places that are likely to be missed by a proofreader if they are unfamiliar with the content):

3: “Satoyama” is not a one-to-one term for the term used here “socio-ecological production landscapes”, but one example of such landscapes in Japan. Would recommend something like “Satoyama, a kind of socio-ecological…”.

42: My reading is that CBD Decision X/32 only “recognizes” and “takes note of” the Satoyama Initiative, but does not “adopt” it.

59: The statement that “In this regard, SEPLs can be considered as a conservation priority” needs to be qualified, as it does not seem that it is currently considered a conservation priority. Something like “SEPLs seem like a good candidate to be a conservation priority” or the like could be better.

62: I think the word “natural” should be “national”.

183: “IPSI” is misspelled here as “ISPI”. An overall check should be done in case this error occurs elsewhere.

451-455: The concept of OECMs is raised here, and then immediately negated in the sentence “Whether as OECM or not…”, making it unclear how OECM is relevant to the argument.

455-456: The percentage of land that should be covered in this kind of global biodiversity policy is a very salient issue in CBD negotiations, and defending the need for 20% seems like it would require much more attention than this casual mention here. It seems beyond the scope of the current paper, unless the authors would like to greatly expand on how the figure of 20% is reached.

481-482: It would be helpful to at least give the headline topics of the SDGs listed here, “Zero Hunger” and “Live on Land”.

484-485: “people-nature coexistence” – the term “nature-social interactions” is used in the abstract. Is this terminology that is or should be made consistent? As a reader I am not sure if it is exactly the same thing. In the same vein, the term “landscapes of people-nature coexistence” is new here. Is it the same as SEPLs?

487-492: This is mentioned earlier in the paper as well and it may be beyond the scope of the current paper, but this seems like the major issue. It is unclear how the essential problem of having data at a sufficient resolution will ever be possible particularly for huge areas of land, and therefore if this kind of research can ever produce the results desired. If there is any more information on the potential of how this can be done practically it would be helpful for the paper.

Reviewer #4: The article is significant in the conservation of SEPLs and can be a useful tool for policy makers. Well researched and supported by data. However, the manuscript should be edited and should be written in standard English since some of the sentences are in first person (e.g. L103, L467). The abstract should also be edited since there are missing information. The abstract should stand alone and when read should provide the summary of the research. Overall, the paper is acceptable.

6. PLOS authors have the option to publish the peer review history of their article (what does this mean?). If published, this will include your full peer review and any attached files.

Reviewer #1: No

Reviewer #2: No

Reviewer #3: No

Reviewer #4: No

---

## [Author Response · Author response to Decision Letter 0]

3 May 2021

Dear Dr. Stephen P. Aldrich, 

Thank you very much for the thorough and thoughtful review of our manuscripts. All points raised by you and reviewers have been addressed in the revised manuscript, as described in the response to reviewers document. We sincerely hope that our manuscript is in order and acceptable this time.

Best regards,

Yoji Natori

---

## [Decision Letter · Decision Letter 1]

5 Aug 2021

Global identification and mapping of socio-ecological production landscapes with the Satoyama Index

PONE-D-20-26225R1

Dear Dr. Natori,

We’re pleased to inform you that your manuscript has been judged scientifically suitable for publication and will be formally accepted for publication once it meets all outstanding technical requirements.

Kind regards,

Stephen P. Aldrich, PhD

Academic Editor

PLOS ONE

Additional Editor Comments (optional):

Reviewers' comments:

Reviewer's Responses to Questions

**Comments to the Author**

1. If the authors have adequately addressed your comments raised in a previous round of review and you feel that this manuscript is now acceptable for publication, you may indicate that here to bypass the “Comments to the Author” section, enter your conflict of interest statement in the “Confidential to Editor” section, and submit your "Accept" recommendation.

Reviewer #1: All comments have been addressed

Reviewer #2: All comments have been addressed

2. Is the manuscript technically sound, and do the data support the conclusions?

Reviewer #1: Yes

Reviewer #2: Yes

3. Has the statistical analysis been performed appropriately and rigorously? 

Reviewer #1: Yes

Reviewer #2: Yes

4. Have the authors made all data underlying the findings in their manuscript fully available?

Reviewer #1: Yes

Reviewer #2: Yes

5. Is the manuscript presented in an intelligible fashion and written in standard English?

Reviewer #1: Yes

Reviewer #2: Yes

6. Review Comments to the Author

Reviewer #1: The comments I had pointed out were thoroughly addressed. I think that the manuscript has become suitable for publication.

Reviewer #2: The manuscript has been significantly improved by the authors in form and content. Greater emphasis is now placed on the adaptations that should be made to the input data for the use of the index. These could also be made in the index itself by means of calibration methods. Nevertheless, the article turns out to be a preliminary analysis of a global identification of SEPL, which is shown that with the data used it is not possible in many regions and perhaps it should be specified in the title so as not to create expectations and in the interest of a future publication of the improved index with greater global applicability.

7. PLOS authors have the option to publish the peer review history of their article (what does this mean?). If published, this will include your full peer review and any attached files.

Reviewer #1: No

Reviewer #2: No

---

## [Editor Report · Acceptance letter]

9 Aug 2021

PONE-D-20-26225R1 

Global identification and mapping of socio-ecological production landscapes with the Satoyama Index 

Dear Dr. Natori:

I'm pleased to inform you that your manuscript has been deemed suitable for publication in PLOS ONE. Congratulations! Your manuscript is now with our production department. 

Kind regards, 

on behalf of

Dr. Stephen P. Aldrich 

Academic Editor

PLOS ONE